# The Global Burden of Meningitis in Children: Challenges with Interpreting Global Health Estimates

**DOI:** 10.3390/microorganisms9020377

**Published:** 2021-02-13

**Authors:** Claire Wright, Natacha Blake, Linda Glennie, Vinny Smith, Rose Bender, Hmwe Kyu, Han Yong Wunrow, Li Liu, Diana Yeung, Maria Deloria Knoll, Brian Wahl, James M. Stuart, Caroline Trotter

**Affiliations:** 1Meningitis Research Foundation, Bristol BS1 5HX, UK; tachablake26@hotmail.co.uk (N.B.); Lindag@meningitis.org (L.G.); vinnys@meningitis.org (V.S.); 2Institute for Health Metrics and Evaluation, University of Washington, Seattle, WA 98105, USA; rbender1@uw.edu (R.B.); hmwekyu@uw.edu (H.K.); hwunrow@uw.edu (H.Y.W.); 3Department of Population, Family and Reproductive Health and Institute for International Programmes, Johns Hopkins Bloomberg School of Public Health, Baltimore, MD 21205, USA; lliu26@jhu.edu; 4Institute for International Programmes, Department of International Health, Johns Hopkins Bloomberg School of Public Health, Baltimore, MD 21205, USA; dyeung@jhu.edu; 5International Vaccine Access Center, Johns Hopkins Bloomberg School of Public Health, Baltimore, MD 21231, USA; mknoll2@jhu.edu (M.D.K.); bwahl@jhu.edu (B.W.); 6Population Health Sciences, Bristol Medical School, University of Bristol, Bristol BS8 1QU, UK; James.Stuart@bristol.ac.uk; 7World Health Organization, 1211 Geneva, Switzerland; 8Disease Dynamics Unit, Department of Veterinary Medicine, University of Cambridge, Cambridge CB3 0ES, UK; clt56@cam.ac.uk

**Keywords:** meningitis, child mortality, neonatal sepsis, global health, global health estimates, modelling, *Streptococcus pneumoniae*, *Haemophilus influenzae*, *Neisseria meningitidis*

## Abstract

The World Health Organization (WHO) has developed a global roadmap to defeat meningitis by 2030. To advocate for and track progress of the roadmap, the burden of meningitis as a syndrome and by pathogen must be accurately defined. Three major global health models estimating meningitis mortality as a syndrome and/or by causative pathogen were identified and compared for the baseline year 2015. Two models, (1) the WHO and the Johns Hopkins Bloomberg School of Public Health’s Maternal and Child Epidemiology Estimation (MCEE) group’s Child Mortality Estimation (WHO-MCEE) and (2) the Institute for Health Metrics and Evaluation (IHME) Global Burden of Disease Study (GBD 2017), identified meningitis, encephalitis and neonatal sepsis, collectively, to be the second and third largest infectious killers of children under five years, respectively. Global meningitis/encephalitis and neonatal sepsis mortality estimates differed more substantially between models than mortality estimates for selected infectious causes of death and all causes of death combined. Estimates at national level and by pathogen also differed markedly between models. Aligning modelled estimates with additional data sources, such as national or sentinel surveillance, could more accurately define the global burden of meningitis and help track progress against the WHO roadmap.

## 1. Introduction

The world saw great progress in reducing child mortality over the lifetime of the United Nations (UN) Millennium Development Goals (MDGs) with an estimated 54% decline in children under five years of age from 93 deaths per 1000 live births in 1990 to 43 per 1000 live births in 2015 [1]. The successor UN Sustainable Development Goals (SDGs) are more ambitious again, and urge that by 2030 we should “end preventable deaths of newborns and children under five years of age, with all countries aiming to reduce neonatal mortality to at least as low as 12 per 1000 live births and under-five mortality to at least as low as 25 per 1000 live births.” However, with the majority of an estimated 38 deaths per 1000 live births in 2019 being caused by preventable and treatable diseases [1], we are a long way from achieving this target.

Among these preventable diseases, meningitis has one of the highest fatality rates and the potential to cause devastating epidemics. Since the turn of the century, we have seen advances as a result of widespread global introduction of *Haemophilus influenzae* type b (Hib) and pneumococcal vaccines as well as the roll out of the meningococcal A vaccine, MenAfriVac, across some of the highest incidence areas of sub-Saharan Africa. Despite this, recent estimates have identified that the global burden of meningitis in all age groups remains high and progress lags substantially behind that of other vaccine preventable diseases [2]. Whilst deaths from measles and tetanus in children under five years are estimated to have decreased by 86% and 92% respectively, between 1990 and 2017, over the same time period deaths from meningitis are estimated to have decreased by just 51% [3]. Despite its burden, meningitis is seldom, if at all, mentioned in key global and regional health documents [4,5,6,7,8,9].

In response to calls from governments, global health organisations, civil society, public health bodies, academia and the private sector, a World Health Organization (WHO)-led collaboration is developing a Defeating Meningitis by 2030 Global Roadmap [10]. The Roadmap focuses on the four leading global causes of bacterial meningitis; *Neisseria meningitidis* (meningococcus), *Streptococcus pneumoniae* (pneumococcus), *Haemophilus influenzae* (Hi), and *Streptococcus agalactiae* (group B streptococcus (GBS)). 

To advocate for a global roadmap to defeat meningitis, the global burden of meningitis as a syndrome in relation to other infectious causes of death needs to be accurately described, and countries with the highest burden identified, so that efforts and resources can be targeted effectively. Estimates of pathogen-specific meningitis incidence and mortality at the global level can identify the need for new vaccines or support wider access to existing ones. Tracking trends in pathogen-specific meningitis and syndromic disease over time at the national and international level is vital to assess the impact of interventions such as vaccines implemented as part of the global roadmap to defeat meningitis. 

Vital registration systems and disease surveillance platforms are limited across many geographies and regions, so there is a reliance on modelled estimates to get a complete global picture of disease across all settings but cause of death estimates have been found to differ across these different modelling efforts [11]. Modelled estimates also attempt to account for changes in causes of death over time, but to do so accurately they must be informed by reliable data to make accurate predictions where real data is lacking. 

In this paper we aim to compare the available modelled estimates for cases and deaths from meningitis as a syndrome, by causative pathogen and the methods used, in order to assess whether these models can be used with confidence by decision makers to prioritise recommendations from a plan to defeat meningitis, and by those needing to track progress on the WHO Defeating Meningitis by 2030 Global Roadmap’.

## 2. Materials and Methods

### 2.1. Identification of Data Sources

Through attending key stakeholder meetings, we identified three modelling efforts that estimate the global burden of meningitis and neonatal sepsis: (1) WHO and the Johns Hopkins Bloomberg School of Public Health’s Maternal and Child Epidemiology Estimation (MCEE) group’s Child Mortality Estimation (WHO-MCEE), which estimates15 causes of death for children under five years of age [12]; (2) The Institute for Health Metrics and Evaluation (IHME) Global Burden of Disease Study (GBD 2017) which estimates age specific mortality for 282 causes of death in all ages [3]; and (3) The WHO’s Global Health Estimates (WHO GHE) which estimates age specific mortality for 136 causes of death in all ages [13]. 

Two additional models were also identified that estimated disease burden caused by pathogens of particular relevance to the WHO Defeating Meningitis by 2030 Global Roadmap: (1) the WHO-MCEE group’s estimates of the burden of pneumococcal and Hib disease in children [14], and (2) the London School of Hygiene and Tropical Medicine (LSHTM) Burden of Group B Streptococcus Worldwide for Pregnant Women, Stillbirths, and Children [15].

Not all of these efforts were directly comparable because they did not provide the same level of data or use the same indicators of burden (Table 1).

As WHO GHE estimates were an amalgamation of historical models (WHO-MCEE’s 2000–2016 and IHME’s GBD 2016) we did not consider them further in our analysis. We did not include GBS estimates from LSHTM in our analysis because the age categorisation (0–89 days) did not correspond with the disaggregated age categories of the other models and so did not allow for meaningful comparison.

### 2.2. Analysis of Data Sources

The scale of the global burden of meningitis deaths relative to all causes and leading infectious causes of death was assessed by comparing, death and mortality estimates from GBD 2017 and the WHO-MCEE’s 2000–2017 model according to the following syndromic cause of death categories “All causes”, “Infectious disease”, “meningitis/encephalitis” and “neonatal sepsis”.

We considered the burden of meningitis and neonatal sepsis together for the purposes of comparison with other leading infectious causes of death because distinguishing between these syndromes is almost impossible based on clinical signs alone in the neonate [16,17]. Lumbar puncture (LP) and analysis of the cerebrospinal fluid is the only reliable way of confirming a case of meningitis. However, in many countries there is a shortage of trained staff to perform LP [18], and in low-income settings as few as 2% of neonates with infection might have an LP or blood sample taken [19].

The WHO-MCEE have historically estimated sepsis and meningitis in the neonatal period within the same cause category because of difficulties in distinguishing between these clinical syndromes in this age group. These causes were estimated separately for the first time in their latest modelling round by using the ratio of neonatal meningitis and neonatal sepsis deaths derived from IHME estimates. Because WHO-MCEE estimate meningitis/encephalitis as one cause category, GBD 2017 meningitis and encephalitis deaths were amalgamated for the purpose of comparison.

Denominators used to report mortality rates were standardised across the models and, where necessary, recalculated to be expressed as deaths per 1000 live births in the neonatal period and deaths per 100,000 population in the post neonatal period. GBD 2017 mortality rates in the neonatal period were calculated from IHME live birth estimates for the year 2015. WHO-MCEE postneonatal mortality rates were calculated using UN population estimates for the year 2015 [20].

Priority geographical areas for targeting a plan to defeat meningitis were identified from country-specific GBD 2017 and WHO-MCEE meningitis/encephalitis mortality estimates for the year 2015 in children under five years.

Meningitis mortality and incidence estimates according to pathogen over time (2000–2015) were analysed using estimates produced by GBD 2017 and the WHO-MCEE pathogen model. Meningococcal meningitis is commonly associated with epidemics. As WHO-MCEE meningococcal meningitis estimates did not account for deaths and cases resulting from epidemics, estimates for ‘Hib meningitis’ and ‘pneumococcal meningitis’ mortality and incidence in the post neonatal period (28 days–<5 years) were the categories and age group used for comparison.

An analysis of the estimation methodology for each model was also undertaken in an attempt to explain any inconsistencies between models.

## 3. Results

### 3.1. Global Meningitis and Neonatal Sepsis Mortality Estimates in Children Aged Under Five Years

Overall, the WHO-MCEE estimated there to be approximately 100,000 fewer deaths in the under-five age group than the GBD 2017, with proportionally more under-five deaths occurring in the neonatal period (46% compared to GBD 2017’s 43%).

The GBD 2017 estimated 34% more deaths from meningitis/encephalitis than the WHO-MCEE in the year 2015 (190,515 and 142,841, respectively) (Table 2). Meningitis made up the majority of the GBD 2017 combined meningitis/encephalitis category; 87% in under five-year-olds, 86% in 1–59 months and 93% in 0–28 days.

However, the WHO-MCEE estimated >100,000 more deaths than the GBD 2017 when neonatal sepsis deaths were combined with meningitis/encephalitis, due to the WHO-MCEE’s much higher estimate of neonatal sepsis deaths. Uncertainty intervals do not overlap between modelled estimates of deaths from neonatal sepsis in any of the age categories. This section may be divided by subheadings. It should provide a concise and precise description of the experimental results, their interpretation, as well as the experimental conclusions that can be drawn.

The WHO-MCEE model estimated meningitis/encephalitis and neonatal sepsis as the second largest infectious cause of death, co-ranked with diarrhoeal diseases, in children aged under five years in 2015, after acute respiratory infections (Figure 1). In contrast the GBD 2017 estimated this cause category to be the third largest infectious case of death after acute respiratory infections and diarrhoeal diseases.

At the country level, there was considerable variability in estimates of burden per population. For meningitis/encephalitis mortality rates, the WHO-MCEE model ranked Somalia highest for the year 2015 (139.7 deaths per 100,000 population), whilst the GBD 2017 ranked Somalia 11th highest for the same year (68.6 deaths per 100,000).

For numbers of deaths, both models attribute approximately 70% of all meningitis/encephalitis deaths in children under five years to just 12 countries including India, Nigeria, Pakistan, Democratic Republic of Congo (DRC), Ethiopia, Niger, Afghanistan, Mali, Uganda and China. However, whilst Somalia and Chad feature in the top 12 (ranked 7th and 8th respectively) in the WHO-MCEE estimates, they did not feature in the GBD 2017 top 12, where Indonesia and Burkina Faso featured instead (ranked 8th and 9th highest respectively) (Figure 2).

### 3.2. Meningitis Incidence and Mortality Estimates by Aetiology in Children Aged Under Five Years

The GBD 2017 and WHO-MCEE’s pathogen models both estimated pneumococcal and Hib meningitis mortality and incidence in children aged 1 to 59 months at the national and global levels.

A comparison of the global estimates for the year 2015 (Table 3) showed that both models agree there were more cases of pneumococcal meningitis than Hib meningitis in 2015. However, whilst the GBD 2017 estimated around twice as many deaths from Hib meningitis compared to pneumococcal meningitis, the WHO-MCEE estimated around five times more deaths from pneumococcal meningitis than from Hib meningitis in the same year.

Despite major differences in the relative proportions of meningitis deaths attributed to Hib and pneumococcal bacteria between models, both models agreed that Hib and pneumococcal meningitis combined were the underlying cause of approximately 40% of all meningitis/encephalitis deaths globally.

When comparisons were made between the modelled estimates for Hib and pneumococcal meningitis incidence and mortality over time (Figure 3), both models showed a steeper decline in Hib meningitis incidence and mortality compared to pneumococcal meningitis mortality, which is consistent with wider roll-out of Hib vaccination globally compared to pneumococcal vaccination. However, the GBD 2017 consistently reported much higher incidence of pneumococcal and Hib meningitis over time compared to the WHO-MCEE. The GBD 2017 estimated Hib and pneumococcal incidence to be 31 and 40 cases per 100,000, respectively, in 2015 compared to the WHO-MCEE estimates of around five and 13 cases per 100,000 for Hib and pneumococcal meningitis, respectively.

Of note is that case fatality rates (CFRs) differed dramatically between the two sets of estimates. CFRs derived from WHO-MCEE global cases, and deaths estimates for Hib and pneumococcal meningitis in 2015, were 23% and 45%, respectively. However, CFRs calculated from GBD 2017 estimates were 8% for pneumococcal meningitis, 19% for Hib meningitis and 8% for meningococcal meningitis. Evidence from the literature closely agrees with the WHO-MCEE CFRs, consistently reporting higher CFRs from pneumococcal meningitis compared to Hib meningitis and meningococcal meningitis [21,22,23,24,25,26].

### 3.3. Modelling Methodology Which Could Account for Differences in Mortality and Pathogen Specific Estimates

Figure 4 depicts a simplified methodology for both modelling approaches. A more detailed explanation is provided in the appendix, and full methodological approaches are also outlined elsewhere [3,27]. When calculating the meningitis death envelope, both models used country-specific death data from vital registration and other sources and applied statistical modelling to fill gaps in the data using country-specific covariates and drawing on trends observed where data was more complete. Whilst the GBD included intervention covariates (such as vaccine coverage) within their cause of death ensemble modelling (CODEm) (Figure 4), the WHO-MCEE model used intervention covariates in both their modelling, and also in post hoc adjustments, to redistribute causes accounting for interventions. Details of the covariates used by the models are available in the Appendix A. Both models ensured that the sum of deaths attributed to different causes fitted within a total all-cause mortality envelope calculated from surveys, censuses and vital registration data.

Whilst there was little difference between estimated mortality from all causes and infectious diseases in children under five years (2% and 4% difference in estimated deaths, respectively), between models there was a marked difference between meningitis/encephalitis and neonatal sepsis mortality estimates in this age group (29 and 53 percent difference, respectively) (Table 2).

Further investigation into the modelling methods and underlying data showed that countries with the highest meningitis burden have the lowest quality death registration data. Whilst this is also the case for all causes of death, a higher proportion of meningitis/encephalitis death estimates were based on extrapolating from low-quality underlying data compared to all-cause death estimates. For example, 77% of meningitis/encephalitis deaths came from countries with no or very low-quality death registration data (scaled 0 to 1) compared to 60% of deaths due to all causes in the GBD 2017 model. Likewise, in the WHO-MCEE model, 95% of meningitis/encephalitis deaths were estimated using modelling underpinned by verbal autopsy (VA) studies compared to 90% of all cause deaths due to these countries having poor quality death registration data (see Appendix A). As would be expected, there were greater differences between estimates from countries with low-quality underlying data compared to those with higher quality data (Figure 5).

To estimate meningitis mortality by aetiology, both models applied a proportional split by pathogen to the country-specific meningitis death envelope and adjusted for vaccine coverage. Whilst GBD 2017 pathogen specific mortality proportions were informed by vital registration (VR) data from data rich locations, the WHO-MCEE model based mortality proportions on studies reporting the distribution of pathogen-specific meningitis cases adjusted by pathogen-specific CFRs to derive proportions of deaths. This approach was used due to a lack of literature reporting meningitis mortality fractions by pathogen. To adjust pathogen-specific estimates according to vaccine coverage, IHME ran a metaregression model (DisMod-MR 2.1) with pneumococcal and Hib vaccine coverage as covariates driving down the proportions of disease attributed to those pathogens. The WHO-MCEE model used a deterministic approach to account for vaccine use by calculating the percentage reduction in disease as a result of vaccine efficacy, coverage and, in the case of PCV, the vaccine product and proportion of disease caused by vaccine-specific serotypes.

The models used very different approaches for estimating incidence by aetiology. The GBD 2017 calculated meningitis incidence independently from meningitis mortality using incidence data gathered from hospital records, claims data and a systematic review of the literature capturing incidence studies. The WHO-MCEE incidence estimates were derived by dividing pathogen-specific death estimates by literature-derived CFRs. The WHO-MCEE also published an update to a previous incidence-based model for Hib and pneumococcal meningitis [28], which predicted even lower incidence rates for pneumococcal meningitis and similar rates for Hib.

## 4. Discussion

Despite major differences in the number of deaths attributed to meningitis, both models agree that there is a substantial burden of disease, with meningitis as either the 2nd or 3rd most important infectious syndrome. By far the biggest burden of meningitis is estimated to occur in countries with low quality or no death registration data where these models rely heavily on extrapolating from VA studies. Accurately attributing meningitis as a cause of death using VA is extremely challenging [29,30,31] and could lead to meningitis as a cause of death being underestimated. VA has a high specificity but low to moderate sensitivity for meningitis [32,33,34] and can easily attribute death from meningitis to a different cause, especially in malaria endemic regions where severe febrile illness is often assumed to be malaria [35,36,37].

If these syndromic models systematically underestimate deaths from meningitis, this would result in an underestimate of incidence by pathogen in the WHO-MCEE model because incidence is derived by dividing estimated deaths by CFR based on location and pathogen. The GBD 2017 estimated pathogen-specific incidence separately to pathogen-specific deaths and produced higher estimates than the WHO-MCEE model, but the incidence estimates were out of line with deaths when literature-derived CFRs were applied. Following a meeting where results from this analysis were presented to all modelling groups, the IHME amended their methodology for calculating pathogen-specific incidence. In the recently published GBD 2019 model [38], published studies and hospital data were used to estimate pathogen-specific CFRs as a function of healthcare access and quality. Pathogen specific mortality was then derived from estimates of pathogen-specific incidence and CFRs.

Using global health estimates to derive baseline numbers and targets against which progress can be measured is challenging. Estimates for the entire time series are updated with successive model iterations as new input data are considered and amendments are made to statistical modelling processes. This means that baseline estimates for a given year fluctuate with successive model iterations.

It is vital that the methods used to derive estimates are clearly communicated. Across models it was unclear from published methods exactly how neonatal meningitis as a cause was disaggregated from neonatal sepsis, and other infectious conditions of the newborn, when we know that the majority of the underlying input data does not distinguish between these two causes of death. Unless methods are made transparent, it is difficult for policy makers to understand, and therefore trust, model outputs [39].

Experts responsible for monitoring progress also need to know exactly how estimates were derived in order to assess whether they are capable of measuring progress against certain indicators. Whilst both models accounted for PCV and Hib vaccine impact, they did so using substantially different methods. The IHME’s GBD 2017 study accounted for vaccine impact by finding existing relationships between vaccine coverage and the proportion of pathogen-specific meningitis targeted by the vaccine (from countries where data is available) and using these existing relationships to make predictions where data is unavailable. Whilst this approach has an advantage of using as much raw data as possible, it does not distinguish between differences in vaccine products and the varying efficacy associated with different dosing schedules between countries. Although incidence proportion models included data from some countries in sub-Saharan Africa and Asia, the use of VR data alone to determine proportional cause of death means that vaccine effects on pathogen-specific mortality in high mortality countries with no vital registration data are heavily reliant on effects demonstrated in data-rich low-mortality countries. The WHO-MCEE, on the other hand, make predictions where data is sparse/unavailable by simulating the effect of a given vaccine over time on a country specific basis. Assumptions about vaccine impact are transparent and take into account differences in vaccine formulations and dosing schedules, but they may be applied to a pathogen specific meningitis death estimate which is highly uncertain.

It is also important for decision makers to be aware that even in data-rich locations, global health estimates for the most recent year can be based on predictions rather than real underlying data. These estimates may, therefore, be unsuitable for tracking change as a result of a recent intervention, especially if the intervention has not been accounted for as a covariate in the model.

The IHME’s GBD model is currently the only available complete source of information about the global and national burden of meningitis amongst all age groups and for most of the pathogens of interest to the global roadmap to defeat meningitis. The IHME have also improved some of their methods for the latest round of estimates by including more surveillance data from high mortality settings in the GBD 2019 than was included in the GBD 2017. Additionally, there are plans for future versions of the IHME’s model to include estimates on the incidence and mortality from GBS meningitis, one of the major causes of meningitis in neonates worldwide. However, tracking outputs from multiple models in parallel has advantages in identifying areas of higher uncertainty, generating opportunities for modellers to improve methods and prioritising further primary data collection/strengthening surveillance. An interactive visualisation has been created to track progress using estimates from all of the major global health estimation models [40].

None of the models we assessed were able to accurately account for the fluctuating scales of periodic, large epidemics of meningitis, which are irregular and unpredictable in nature. Whilst GBD 2017 attempted to account for epidemic meningococcal meningitis deaths by adding these to the meningitis death envelope, they did not use equivalent methodology to account for epidemic meningococcal meningitis cases. The WHO-MCEE syndromic model attempted to account for epidemic disease by estimating the average increase in deaths in epidemic years relative to nonepidemic years and adding these to estimates in years with epidemics identified by WHO surveillance reports and published literature. This increases estimates during an epidemic year, but the underlying data from the country are not always reliable, and it does not accurately reflect the variation in the size of the epidemic for a given year. The WHO-MCEE pathogenic model only estimated pathogen-specific deaths for endemic disease, removing the simulated effects of epidemics from the syndromic model before applying proportional splits to the remaining meningitis envelope. Therefore, neither model estimating pathogen specific causes of meningitis was able to account for epidemic pneumococcal meningitis, yet this is an important consideration because it has been demonstrated as having a significant mortality burden [41].

Considering the current limitations of modelled meningitis estimates, it is desirable to track progress alongside additional data where possible. Countries across the African meningitis belt experience the highest burden of meningitis globally because they are susceptible to large and devastating outbreaks of meningococcal disease linked to climatic factors such as dry winds, low humidity and high levels of dust in the air [42]. Whilst many of these countries have poor death registration systems, they have relatively rich and complementary meningitis surveillance systems. Since 2003 an enhanced meningitis surveillance network has been established across the meningitis belt to strengthen outbreak detection and enable a rapid response to outbreaks of meningococcal disease across the region [43]. The network now covers 24 countries, reporting suspected cases and deaths from meningitis to the WHO intercountry support team (WHO/IST) each week during the meningitis season and every month for the rest of the year [44]. Case-based surveillance systems have been established in five countries within the region allowing for comprehensive information on CFRs by age [45].

Triangulating modelled estimates against surveillance data provides the opportunity to reality-check modelled outputs. Utilising surveillance data in combination with evidence of age and regionally specific CFRs has already successfully been used by experts wishing to monitor global progress towards the 2005 measles mortality reduction goal because measles mortality estimates calculated from vital registration data were considered an unreliable way to track progress [46]. Surveillance data for meningitis is not currently available for every country worldwide. However, comprehensive roll out of Hib and pneumococcal vaccines is driving down incidence and mortality from meningitis caused by these pathogens across the globe. Improved pathogen-specific surveillance informed by accurate and timely laboratory diagnosis is required to adequately assess the impact of these important life-saving interventions. This is particularly important for countries transitioning out of Gavi support which need to justify national investments in these vaccines. Additionally, all member states of the UN have committed to achieving universal health coverage by signing up to the SDGs, so there is reason to believe that the availability of good quality surveillance data will improve over time as health systems are strengthened.

More work is required to provide credible meningitis burden estimates for measuring progress. Currently meningitis mortality estimates are highly uncertain because the models rely heavily on death registration data, which is largely missing or incomplete in countries with the highest meningitis burden. Additionally, since postmortem examination is rarely performed in countries without vital registration systems, and the symptoms of meningitis can easily be mistaken for other diseases, there is a risk that the mortality burden of meningitis could be underestimated. Encouragingly, better data on cause of death are becoming available in regions where child mortality rates are the highest through the use of minimally invasive tissue sampling [47,48] and inclusion of these data in future models could considerably improve the reliability of their outputs.

## 5. Conclusions

Global meningitis estimates should be interpreted with caution. Tracking progress towards controlling this disease should also include analysis of real surveillance data where available. The WHO Defeating Meningitis by 2030 Global Roadmap will improve awareness, diagnosis and surveillance of meningitis. As the roadmap drives more comprehensive data on meningitis, a convergence in modelled estimates and a more reliable picture of reductions in the burden of meningitis are anticipated.

## Figures and Tables

**Figure 1 microorganisms-09-00377-f001:**
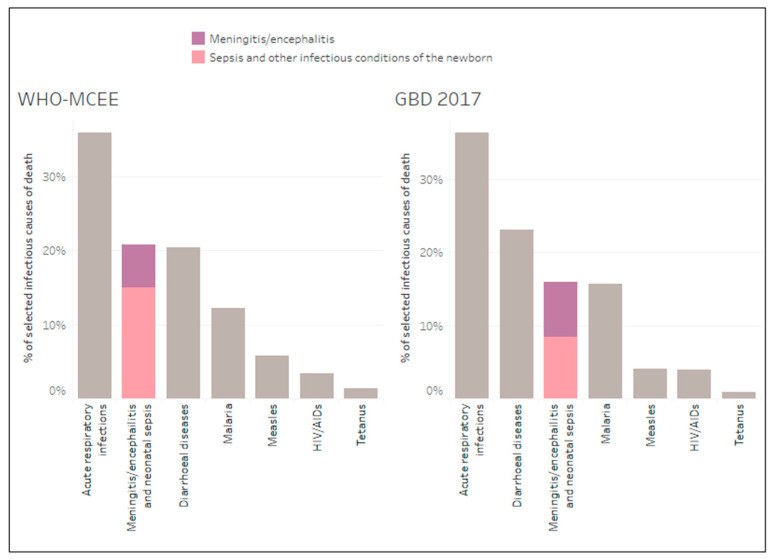
Meningitis/encephalitis and neonatal sepsis mortality burden estimates by model in relation to other selected infectious causes of death in children under five for the year 2015.

**Figure 2 microorganisms-09-00377-f002:**
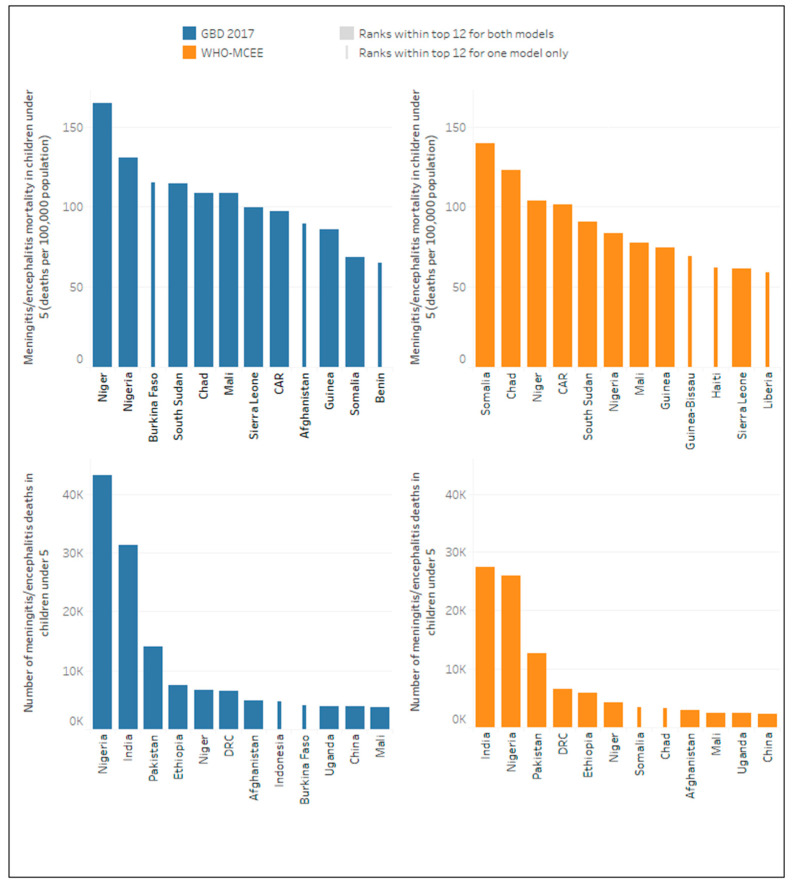
Top twelve ranking countries by meningitis/encephalitis mortality (number and rate) according to model for the year 2015.

**Figure 3 microorganisms-09-00377-f003:**
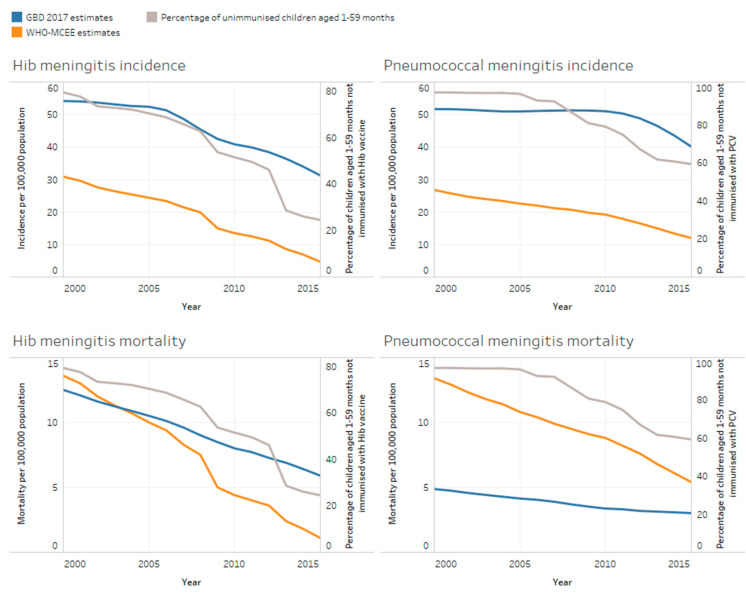
Estimated Hib/pneumococcal meningitis mortality and incidence amongst children aged 1–59 months according to model in relation to the proportion of children unimmunised with Hib vaccine and pneumococcal conjugate vaccine (PCV) over time.

**Figure 4 microorganisms-09-00377-f004:**
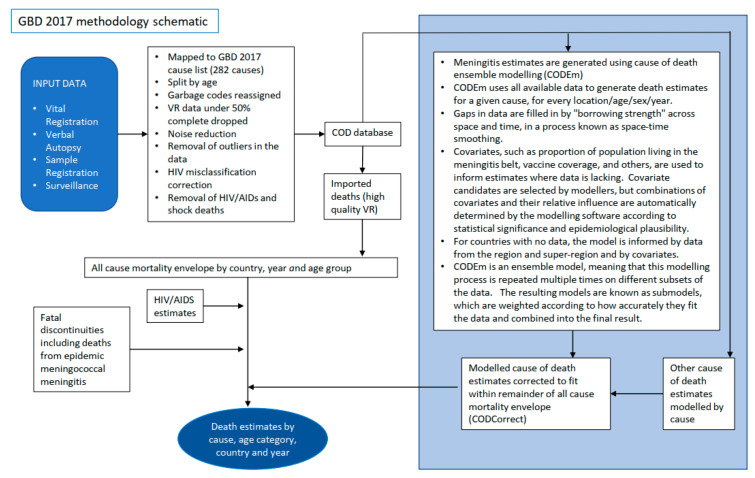
Simplified schematic of the different mortality modelling approaches. VR Data—Data from 76 countries with high quality VR data covering >80% of the population was mapped directly to cause of death categories (see appendix for ICD10 codes mapped to meningitis and sepsis and other severe infections in the neonatal period). VRMCM—Data from the countries with high quality VR data was used to fit a multinomial logistic regression model which was used to predict cause of death proportions in 38 low mortality countries (<35 deaths/1000 live births 2000–2010) with low quality VR data. Covariates used in the model are provided in appendix. VAMCM—In 78 high mortality countries (>35 deaths/1000 live births 2000–2010) verbal autopsy data from 119 research studies in 39 high mortality countries was used to fit a multinomial model to predict causes of death. Cause of death proportions for India were estimated using a combination of VAMCM for the neonatal period and data from the million deaths study and INDEPTH sites in India for the post neonatal period. See appendix for model covariates Other – Cause of death proportions for China were estimated using data from the China Maternal and Child Health Surveillance system. A complete explanation of methods used to produce WHO/MCEE estimates is outlined elsewhere [27].

**Figure 5 microorganisms-09-00377-f005:**
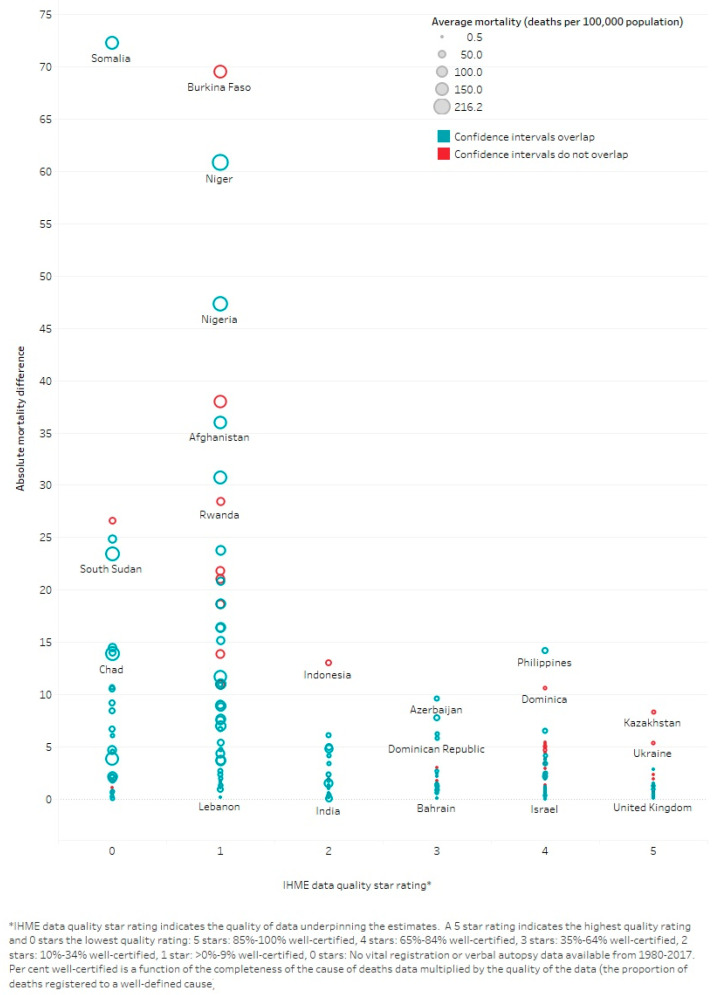
Absolute difference between WHO-MCEE and GBD 2017 meningitis/encephalitis mortality estimates according to country.

**Table 1 microorganisms-09-00377-t001:** Models estimating the global burden of meningitis and neonatal sepsis.

	GBD 2017	WHO GHE	WHO-MCEE Syndromic Model	WHO-MCEE Pathogen Model	LSHTM
Years	1990–2017	2000–2016	2000–2017	2000–2015	2015
Number of countries & territories	195	183	194	194	195
Global under five population estimate in 2015	678,053,340	673,253,870 **	671,355,776 **	657,127,399 ***	N/A
Age range	All ages (Including: Early neonatal: 0–6 days Late neonatal: 7–27 days Post neonatal: 28–364 days 1–4 years)	All ages (including: 0–28 days 1–59 months)	0–59 months (including: 0–28 days 1–59 months)	1–59 months	0–89 days
Relevant disease categories	Meningitis, neonatal sepsis and other neonatal infections	Meningitis *, neonatal sepsis and infections	Meningitis/encephalitis, sepsis and other infectious conditions of the newborn	Meningitis, Non-pneumonia/non-meningitis (which is primarily but not exclusively sepsis)	Meningitis, Sepsis
Outputs	Cases, Incidence rate, Prevalence, Deaths, Mortality rate, DALYs	Deaths, Mortality rate, DALYs	Deaths, Mortality rate	Cases, Incidence rate, Deaths, Mortality rate	Cases, Incidence rate, Deaths, Mortality rate
Published rate per population	Per 100,000 population	Per 100,000 population	Per 1000 livebirths	Per 100,000 population	Per 1000 livebirths
Aetiology	Nm, Spn, Hib, Other	No breakdown by aetiology	No breakdown by aetiology	Nonepidemic disease from: Spn, Hib, Nm	GBS

DALYs = Disability Adjusted Life Years; GBS = Group B streptococcus; Hib = *Haemophilus influenzae* type b; Nm = *Neisseria meningitidis* (meningococcus); Spn = *Streptococcus pneumoniae* (pneumococcus).* WHO GHE use a ratio of meningitis to encephalitis deaths obtained from IHME data to separate out MCEE under-five meningitis/encephalitis estimates. ** Estimates derived from UN World Population Prospects 2017. Differences between WHO GHE and WHO-MCEE population estimates likely due to draft estimates circulating prior to final publication. *** Derived from UN World Population Prospects 2015

**Table 2 microorganisms-09-00377-t002:** Estimated deaths by cause and model for the year 2015 in children under five years of age.

		GBD 2017		WHO-MCEE Pathogen Model		Difference *
		n	Rate ^†^	n	Rate ^†^	%
All causes	Under 5	5,917,285	872.69	5,792,509	862.81 a	2%
		(5,723,776–6,120,099)	(844.15–902.60)	(5,573,633–6,123,477)		
	1–59 months	3,354,404	502.51	3,122,698	473.02 a	7%
		(3,231,491–3,483,015)	(484.10–521.78)	(2,700,899–3,581,030		
	0–28 days	2,562,881	18.40	2,669,811	19.01	−4%
		(2,478,272–2,655,261)	(17.20–19.58)	(2,542,447–2,872,734)	(18.10–20.50)	
Infectious diseases **	Under 5	2,519,567	371.59	2,426,882	361.49 a	4%
		(2,379,024–2,671,856)	(350.86–394.05)	(2,279,602–3,169,783)		
	1–59 months	1,967,826	294.79	1,810,771	274.29 a	8%
		(1,847,763–2,091,762)	(276.81–313.36)	(1,703,587–2,350,572)		
	0–28 days	551,740	3.96	616,111	4.39	−11%
		(510,918–603,527	(3.60–4.38)	(605,290–877,610)	(4.31–6.25)	
Meningitis & Encephalitis	Under 5	190,515	28.10	142,841	21.28 a	29%
		(163,374–217,259)	(24.09–32.04)	(87,427–178,552)		
	1–59 months	167,880	25.15	105,406	15.97 a	46%
		(143,529–192,447)	(21.50–28.83)	(87,188–145,213)		
	0–28 days	22,636	0.16	37,435	0.27	−49%
		(18,532–25,642)	(0.13–0.19)	(157–51,299)	(0.001–0.37)	
Neonatal sepsis	Under 5	211,273	31.16	364,188	54.25 a	−53%
		(186,657–275,821)	(27.53–40.68)	(282,744–524,021)		
	1–59 months	12,693	1.90	386 b	0.06 a	188%
		(10,626–16,586)	(1.59–2.48)	(14–579)		
	0–28 days	198,580	1.43	363,802	2.59	−59%
		(175,866–263,096)	(1.24–1.86)	(282,341–523,853)	(2.01–3.73)	

* Percent difference (n) = (GBD 2017–WHO-MCEE)/((GBD 2017 + WHO-MCEE)/2) × 100. ** Sum of specific infectious diseases from WHO-MCEE cause list (HIV/AIDS; diarrhoeal diseases; tetanus; measles; meningitis/encephalitis; malaria; acute respiratory infections; sepsis and other infectious conditions of the newborn). † Rates per 100,000 population in ‘Under 5′ and ‘1–59 months’, and per 1000 livebirths for ‘0–28 days’. a Uncertainty intervals not available–rate calculated using n and under-5 population statistic from UN WPP 2017 Revision–year 2015 (1–59 months calculated using 59/60 months population). b Figures only account for neonatal sepsis deaths in China.

**Table 3 microorganisms-09-00377-t003:** Global aetiology-specific meningitis deaths and cases, 2015, in children aged 1–59 months.

		GBD 2017		WHO-MCEE Pathogen Model		Difference *
		n	Rate ^†^	n	Rate ^†^	%
Pneumococcal meningitis	Cases	267,686	40.10	83,809	13	105%
		(179,314–374,902)	(26.86–56.16)	(36,160–168,500)	(5–26)	
	Deaths	20,156	3.02	37,964	5	−61%
		(16,114–25,199)	(2.41–3.78)	(15,397–79,718)	(2–11)	
Hib meningitis	Cases	208,658	31.26	31,243	5	148%
		(139,815–304,035)	(20.95–45.55)	(13,386–50,595)	(2–8)	
	Deaths	39,380	5.90	7156	1	138%
		(31,782–48,754)	(4.76–7.30)	(2707–11,320)	(0–2)	

* Percent difference = (GBD 2017–WHO-MCEE)/((GBD 2017 + WHO-MCEE)/2) × 100. † Rates per 100,000 population

## Data Availability

Publicly available datasets were analyzed in this study. GBD 2017 estimates are available from http://ghdx.healthdata.org/gbd-2017; WHO-MCEE syndromic meningitis estimates are available from http://158.232.12.119/healthinfo/global_burden_disease/estimates/en/index2.html and WHO-MCEE syndromic estimates are available from their publication [14].

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
