# Peer review of "The Global Burden of Meningitis in Children: Challenges with Interpreting Global Health Estimates"

_microorganisms, 2021, doi:10.3390/microorganisms9020377_

Round 1

Reviewer 1 Report

Authors of this manuscript evaluated the data of global meningitis incidence and mortality obtained by the global health models for past 20 or 30 years. They found that there were substantial differences between the models used and concluded that additional data sources such as national or sentinel surveillance are needed to accurately define the global burden of meningitis. The study design was proper, and the information is important for the field. The manuscript was clearly written.

It is well known that immunisations against causative agents of meningitis have an important impact on incidence and mortality, and that accurate and timely laboratory diagnosis is a basis for surveillance, especially for pathogen specific surveillance. The authors should briefly discuss their importance.

Author Response

We thank the reviewer for their comments and positive feedback. We have now included some wording pointing out the importance of vaccines as life-saving interventions, commenting on how lab-based surveillance is vital to assess their impact and will help drive better data on meningitis burden.

We have amended the penultimate paragraph within the discussion as follows:

Triangulating modelled estimates against surveillance data provides the opportunity to reality-check modelled outputs. Utilising surveillance data in combination with evidence of age and regionally specific CFRs has already successfully been used by experts wishing to monitor global progress towards the 2005 measles mortality reduction goal because measles mortality estimates calculated from vital registration data were considered an unreliable way to track progress [46]. Surveillance data for meningitis is not currently available for every country globe worldwide. However, comprehensive roll out of Hib and pneumococcal vaccines is driving down incidence and mortality from meningitis caused by these pathogens across the globe. Improved pathogen- specific surveillance informed by accurate and timely laboratory diagnosis is required to adequately assess the impact of these important life-saving interventions. This is particularly important for countries transitioning out of Gavi support which need to justify national investments in these vaccines. Additionally. However, all member states of the UN have committed to achieving universal health coverage by signing up to the SDGs so there is reason to believe that the availability of good quality surveillance data will improve over time as health systems are strengthened

Reviewer 2 Report

The global burden of meningitis in children still continues to remain a global challenge. This is an interesting article with focus on its application to extended things. 

Author Response

We thank the reviewer for this positive feedback.

Reviewer 3 Report

The article entitled " The global burden of Meningitis in children" presents  in a scientific manner the impact of  different statistical Tools on achievement of the goal of WHO to defeat the meningitis by 2030. The article is well written and presents  also ideas of further development und improvment of such Tools. Therefore I consider that article should be published.

Author Response

(The authors gave the same response as above.)

Reviewer 4 Report

Dear Authors, 

I would like to start congratulating you for the article. I consider it well-written and very informative. The title is in accordance with the article and gives the readers a clear idea f what the target and research is about. 

Abstract, in my opinion, should include material and method and a short conclusion. The introduction describes the global burden of meningitis in children and child mortality. In order to reduce mortality the authors describe and analyze models that may help track progress. 

Introduction is consistent with information and you describe very carefully the three modelling efforts that estimate the global burden of meningitis and neonatal sepsis and also, two additional models. I do not have any recommendation for this section. 

Material and method section was clear and easy to understand. It consist of two section: identification and analysis of data sources. The result included good and helpful figures. 

Discussion section included the flaws of meningitis mortality models and we agree that more work needs to be done is order to reduce the burden of meningitis. Conclusion is simple and summarises the findings. I salute the effort of the World Health Organisation for reducing the burden of meningitis by creating the road map to defeat it. 

Author Response

We thank the reviewer for their comments and positive feedback.

With regards to the suggestion about the abstract, the journal Microorganisms stipulates that the abstract should be a single paragraph and should follow the style of structured abstracts, but without headings. It also stipulates that the word limit for the abstract is 200 words. A brief description of the methods is contained within the current abstract “Three major global health models estimating meningitis mortality as a syndrome and/or by causative pathogen were identified and compared for the baseline year 2015” and we conclude “Aligning modelled estimates with additional data sources such as national or sentinel surveillance could more accurately define the global burden of meningitis, and help track progress against the WHO roadmap”. As we are quite heavily restricted by the word limit we are unable to elaborate much more on these sections.